# Genetic regulatory effects modified by immune activation contribute to autoimmune disease associations

Sarah Kim-Hellmuth [1,2,3,4], Matthias Bechheim[5], Benno Pütz[6], Pejman Mohammadi [1,2], Yohann Nédélec[7,8], Nicholas Giangreco[2], Jessica Becker[3,4], Vera Kaiser[5], Nadine Fricker[3,4], Esther Beier[5], Peter Boor [9], Stephane E. Castel[1,2], Markus M. Nöthen[3,4], Luis B. Barreiro[7,10], Joseph K. Pickrell[1,11], Bertram Müller-Myhsok[6,12,13], Tuuli Lappalainen[1,2], Johannes Schumacher[3,4] & Veit Hornung[5,14,15]

The immune system plays a major role in human health and disease, and understanding genetic causes of interindividual variability of immune responses is vital. Here, we isolate monocytes from 134 genotyped individuals, stimulate these cells with three defined microbe-associated molecular patterns (LPS, MDP, and 5′-ppp-dsRNA), and profile the transcriptomes at three time points. Mapping expression quantitative trait loci (eQTL), we identify 417 response eQTLs (reQTLs) with varying effects between conditions. We characterize the dynamics of genetic regulation on early and late immune response and observe an enrichment of reQTLs in distal *cis*-regulatory elements. In addition, reQTLs are enriched for recent positive selection with an evolutionary trend towards enhanced immune response. Finally, we uncover reQTL effects in multiple GWAS loci and show a stronger enrichment for response than constant eQTLs in GWAS signals of several autoimmune diseases. This demonstrates the importance of infectious stimuli in modifying genetic predisposition to disease.

[1] New York Genome Center, New York, NY 10013, USA. [2] Department of Systems Biology, Columbia University, New York, NY 10032, USA. [3] Institute of Human Genetics, University of Bonn, Bonn 53127, Germany. [4] Department of Genomics, Life & Brain Center, University of Bonn, Bonn 53127, Germany. [5] Institute of Molecular Medicine, University of Bonn, Bonn 53127, Germany. [6] Statistical Genetics, Max Planck Institute of Psychiatry, Munich 80804, Germany. [7] Department of Genetics, CHU Sainte-Justine Research Center, Montreal Canada H3T 1C5. [8] Department of Biochemistry, University of Montreal, Montreal Canada H3C 3J7. [9] Institute of Pathology and Department of Nephrology, University Clinic of RWTH Aachen, Aachen 52074, Germany. [10] Department of Pediatrics, University of Montreal, Montreal Canada H3T 1C5. [11] Department of Biological Sciences, Columbia University, New York NY 10027, USA. [12] Munich Cluster for Systems Neurology (SyNergy), Munich 80804, Germany. [13] Institute of Translational Medicine, University of Liverpool, Liverpool L69 3GL, UK. [14] Gene Center and Department of Biochemistry, Ludwig-Maximilians-Universität Munich, Munich 81377, Germany. [15] Center for Integrated Protein Science (CIPSM), Ludwig-Maximilians-Universität Munich, Munich 81377, Germany. Tuuli Lappalainen, Johannes Schumacher and Veit Hornung contributed equally to this work   Correspondence and requests for materials should be addressed to S.K.-H. (email: skim@nygenome.org) or to T.L. (email: tlappalainen@nygenome.org) or to J.S. (email: johannes.schumacher@uni-bonn.de)

The human immune system plays an important role in host protection, autoimmune and inflammatory diseases, cancer, metabolism, and ageing. Given this central role in many human pathologies, it is crucial to understand the variability of immune responses at the population level and how this variability relates to disease susceptibility. Studying the genetic influence on immune response is complicated by the complexity of the immune system, which consists of many different cell types that respond to a plethora of signals, interact with each other and induce different effector functions under diverse kinetics[1–5].

An increasingly popular approach to identifying genetic factors influencing the interindividual variation in immune response is to map expression quantitative trait loci (eQTLs) —variants that associate with gene expression—and to identify so-called response eQTLs (reQTLs) where the eQTL effect differs between immune stimuli. Such genetic variants can impact the transcriptional response to infection, and also represent genetic effects that are modified by the infectious environment via gene-by-environment interactions. We and other groups have previously published reQTL studies of stimulated immune cells and demonstrated that the effects of a genetic variant on gene expression are highly context-specific and informative for disease[6–11]. However, given the complexity of the immune system, additional work is needed to illuminate the genetic influence on many aspects of the immune system. For instance, reQTLs of certain pattern recognition receptor (PRR) families such as NOD-like receptors using purified microbial ligands have not been studied yet, and thus far the dynamics of immune reQTLs have only been explored in LPS-treated cells[9].

Building on our previous study of baseline and LPS-stimulated monocytes[6], we address these gaps by studying functional genetic variants in monocytes activated with microbial ligands for three different PRR families at two different time points. We identify context-specific reQTLs and describe their specificity for time point and treatment. In addition, we analyze differences in reQTLs and constant eQTLs in terms of their genetic architecture and contribution to explain GWAS loci. Finally, we describe reQTLs that shed light on the pathogenesis of immune-mediated diseases. Collectively, these results improve our understanding of the complexity of genetic regulation of the immune system. We provide a user-friendly access to our results via the ImmunPop QTL browser (http://immunpop.com/kim/eQTL).

## Results

**Expression profiling of innate immune responses.** To examine the time course of innate immune responses, we first profiled gene expression in monocytes of five individuals using Human HT-12 v4 Expression BeadChips (Illumina) at six time points after stimulation with three prototypical microbial ligands: Lipopolysaccharide (LPS) was used to activate Toll-like receptor 4 (TLR4), muramyl-dipeptide (MDP) to stimulate Nucleotide-binding oligomerization domain-containing protein 2 (NOD2), and 5′-triphosphate RNA (RNA) to activate retinoic acid-inducible gene I (RIG-I). Hierarchical clustering revealed early differentially expressed (DE) genes at 45 and 90 min after stimulation and late DE genes between 3 and 24 h (Supplementary Fig. 1). For the full eQTL cohort, we analyzed primary monocytes isolated from 134 healthy male individuals (185 before quality control), which were either left untreated (baseline) or stimulated with the same three pathogen-derived stimuli, and gene expression was profiled after 90 min and 6 h. All donors were SNP-genotyped using Illumina HumanOmniExpress BeadChips (Fig. 1a). In a previous study[6], we have analyzed a subset of these data consisting of baseline and 90 min LPS-stimulated monocytes in this cohort.

First, we studied the gene expression response to immune stimulation. Principal component analysis of the gene expression data identified seven distinct groups corresponding to each treatment and time point (Supplementary Fig. 2). Differential expression analysis of genes expressed in at least one of the seven conditions showed the highest number of DE genes under late LPS response, and lowest under early RNA stimulation (Supplementary Fig. 3, Supplementary Data 1). These genes form six clusters with similar response patterns across time points and conditions (Fig. 1b, Supplementary Data 1), and with gene ontology (GO) enrichments corresponding to relevant immunological pathways (Supplementary Data 1). Furthermore, immune responsive genes showed a significantly greater and more diverse distribution of interindividual variance than all expressed genes, already at baseline and with a further increase upon stimulation (Supplementary Fig. 4). These analyses of gene expression patterns in a population scale provide a highly robust and comprehensive data set of innate immune responses and their interindividual variation upon diverse microbial ligands and multiple time points.

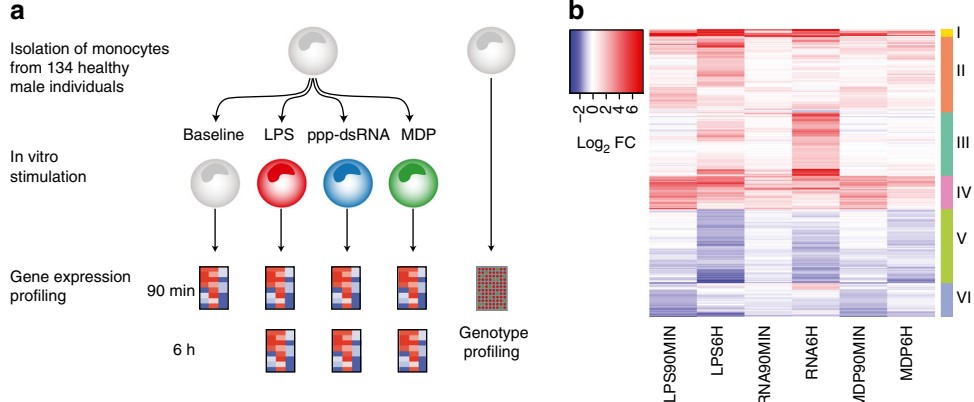

**Fig. 1** Overview of the eQTL study and transcriptional immune response in primary human monocytes. **a** Step-wise experimental design to identify genetic effects on immune response in human monocytes. (1) Isolation and stimulation of primary monocytes from 134 individuals, (2) Transcriptome measurement of the entire cohort at two time points (90 min and 6 h) after stimulation, (3) Genotype profiling to map immune response eQTLs. **b** Mean mRNA profiles of differentially expressed genes (log₂-fold change > 1, FDR 0.001) of 134 individuals between baseline and each of the six stimulated conditions. Genes are hierarchically clustered into six distinct expression patterns (Supplementary Data 1 for a full list of the differential expression and enriched pathways of each cluster)

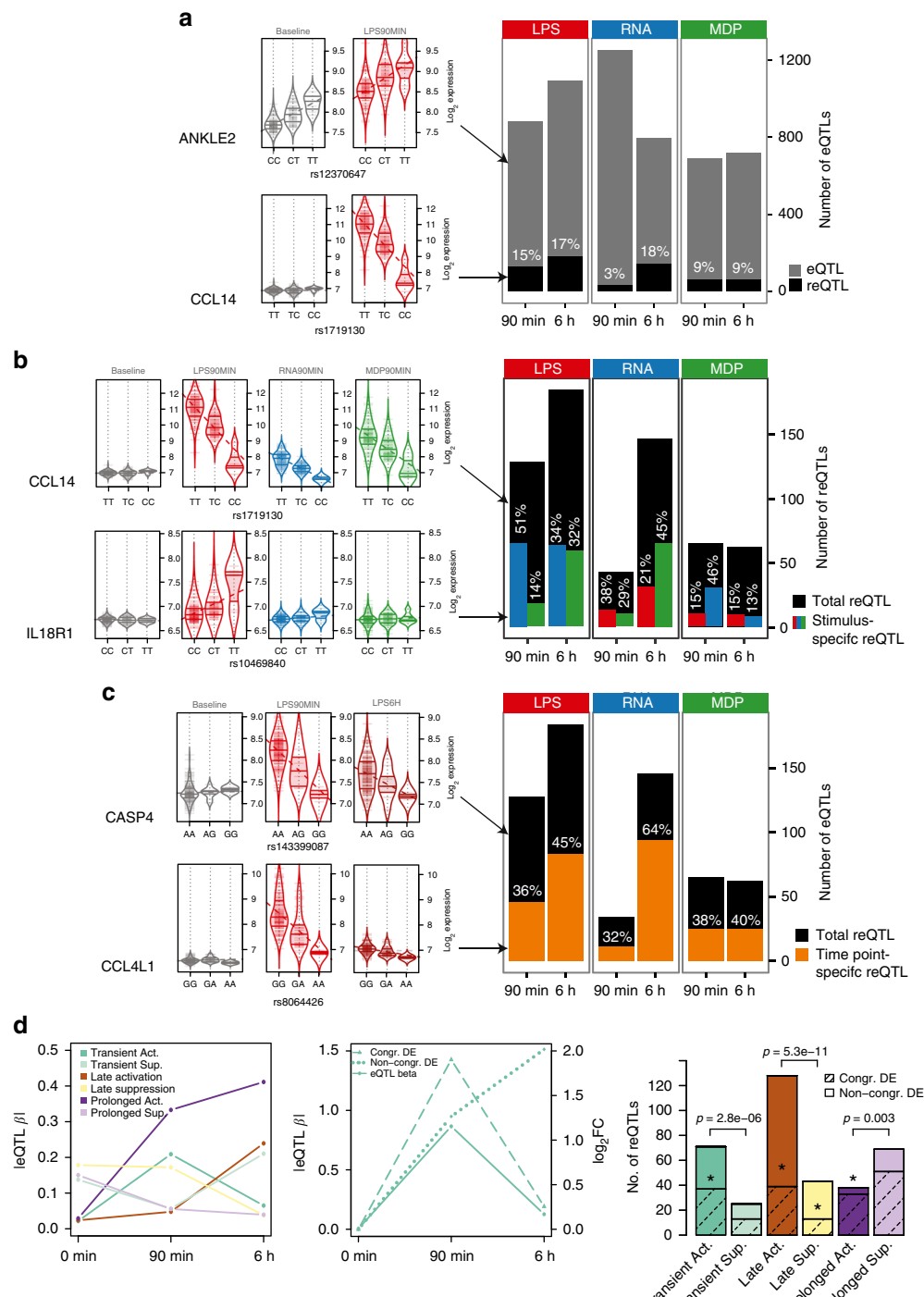

**Fig. 2** Immune response eQTL study in human monocytes. **a** Total numbers of *cis* eQTLs and proportions of reQTLs of LPS-treated (LPS), 5′-ppp-dsRNA (RNA) and MDP-treated (MDP) monocytes at 90 min and 6 h after stimulation. Results of the analysis of 134 individuals are shown unless indicated otherwise. eQTLs include all genes with a significant genetic association in each stimulated condition, and reQTLs are a subset that show a significant difference of the regression slope between untreated and stimulated monocytes, with *violin plots* shown as examples. The untreated condition has 1653 eQTLs that are not shown in the *bar plot*. **b** Numbers of reQTLs and proportions of treatment-specific reQTLs where the regression slope of the tested treatment is different from the slope of the other two treatments within the same time point, with *violin plots* shown as examples and the color of bar indicating the treatment that was tested. **c** Numbers of reQTLs and proportions of time point-specific reQTLs where the regression slope of the tested time point is different from the slope of the other time point within the same treatment, with *violin plots* shown as examples. **d** reQTLs were divided into six subsets according to their temporal activity (see Methods section). Average of absolute eQTL effect sizes per category is shown on the *left panel*. The *middle panel* illustrates a reQTL example with congruent differential expression (DE) (*dashed line*) or non-congruent DE (*dotted line*) of the eGene. reQTL distribution to different categories is shown in the *right panel*, where the *shaded portion* illustrates the proportion of reQTLs with congruent DE of the eGene and *asterisks* represent the significance of enrichment of reQTLs with congruent DE of the eGene (Fisher's exact test *p<0.05). The *p*-values above the *bars* indicate the significance between of active and suppressive types (binomial test)

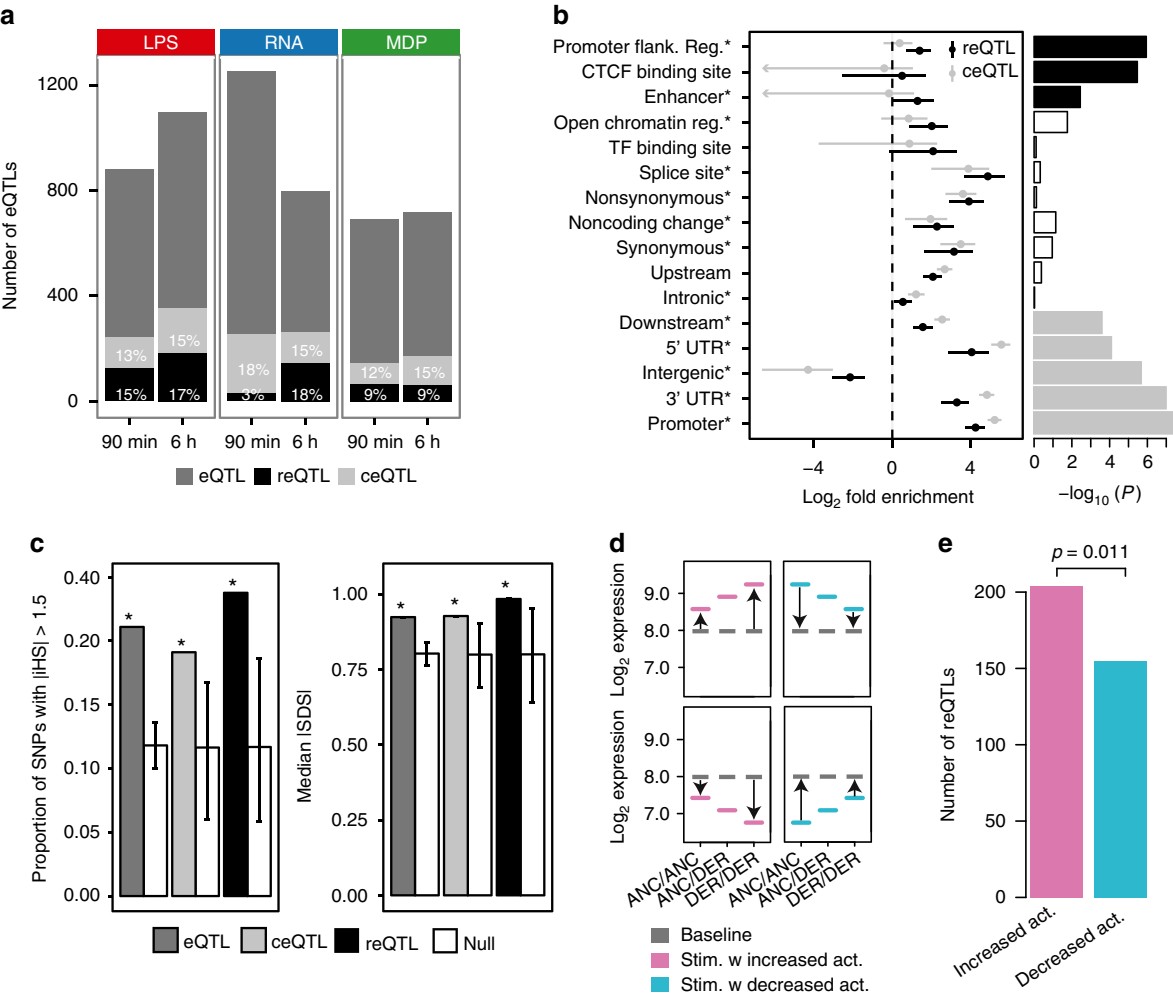

**Fig. 3** Functional annotations and signs of natural selection in reQTLs. **a** Total numbers of *cis* eQTLs, and proportions of reQTLs and constant eQTLs (ceQTL) that have similar regression slopes across all conditions. Results of the analysis of 134 individuals are shown unless indicated otherwise. Examples of a ceQTL and reQTL are shown in Supplementary Fig. 8a. **b** *Forest plot* of enrichment estimates of reQTL and ceQTL signals for each functional annotation with 95% confidence intervals (see also Supplementary Fig. 5b). *Asterisks* indicate annotations that improved the model likelihood in a stepwise procedure for the final best-fitting model. *Bar plot* shows the enrichment of the single most likely causal SNP per locus after fine mapping. The *solid bars* indicate significant enrichments after Bonferroni correction. **c** Signal of positive selection measured as the proportion of variants with high |iHSI| (*left panel*), and median |SDSI| (*right panel*), using the variant with the maximum value from each locus across all SNPs in high LD ($r^2 > 0.8$). Genome-wide null sets of variants matched to eQTL, ceQTL or reQTL were generated by resampling 10,000 sets of random SNPs that matched for MAF and LD (*white bars*). *Error bars* indicate minimum and maximum of the null distribution, and *asterisks* indicate the significant enrichment compared to the null (permutation test $p < 10^{-4}$). **d** Illustration of reQTLs where the derived allele causes an increase (*left panel*) or decrease (*right panel*) in response amplitude compared to the ancestral allele. The increase or decrease of the response amplitude can be in both directions, e.g., reQTLs that amplify the induction or amplify the suppression of a gene are both considered as reQTLs with "increasing activity" of the derived allele and reQTLs that weaken the induction or suppression of a gene are both considered as reQTLs with "decreasing activity" of the derived allele. **e** Numbers of reQTLs with increased or decreased activity across all stimulated conditions, with a *p*-value from a binomial test

**Dynamics of immune response eQTLs**. In order to study genetic variation influencing gene expression levels, we performed eQTL mapping for all seven conditions, defining *cis* eQTLs within a 1-Mb interval on either side of an expression probe at a false discovery rate (FDR) of 5%. We identified 717–1653 genes with an eQTL in each condition (Fig. 2a, Supplementary Data 2). The eQTLs from conditions analyzed in previous studies[8, 9] had a high degree of replication, demonstrating the robustness of our data set (Supplementary Fig. 5a; Methods). We provide a user-friendly access to our results via the ImmunPop QTL browser (http://immunpop.com/kim/eQTL).

To identify eQTLs that differ between stimuli, we used a beta-comparison approach, comparing the regression slopes of

an eQTL under baseline ($\beta_{baseline}$) vs stimulated (e.g., $\beta_{LPS90min}$) in a *z*-test, with reQTLs defined as having Bonferroni corrected $p < 0.05$ (see Methods section). This approach is highly consistent with a previously used method[6, 8, 10] where differential expression is used as the quantitative trait (Supplementary Fig. 5b), but provides more flexibility for comparing several conditions. This analysis revealed that 3–18% of our *cis* eQTLs in each condition are reQTLs (Fig. 2a, Supplementary Data 2). Of note, reQTLs with clearly opposite directional effect when comparing different treatment conditions were not observed (Supplementary Fig. 5c). Genes with a reQTL showed GO enrichment in immune pathways (Supplementary Fig. 6a), and include key genes of protein–protein interaction networks such as MAP kinases,

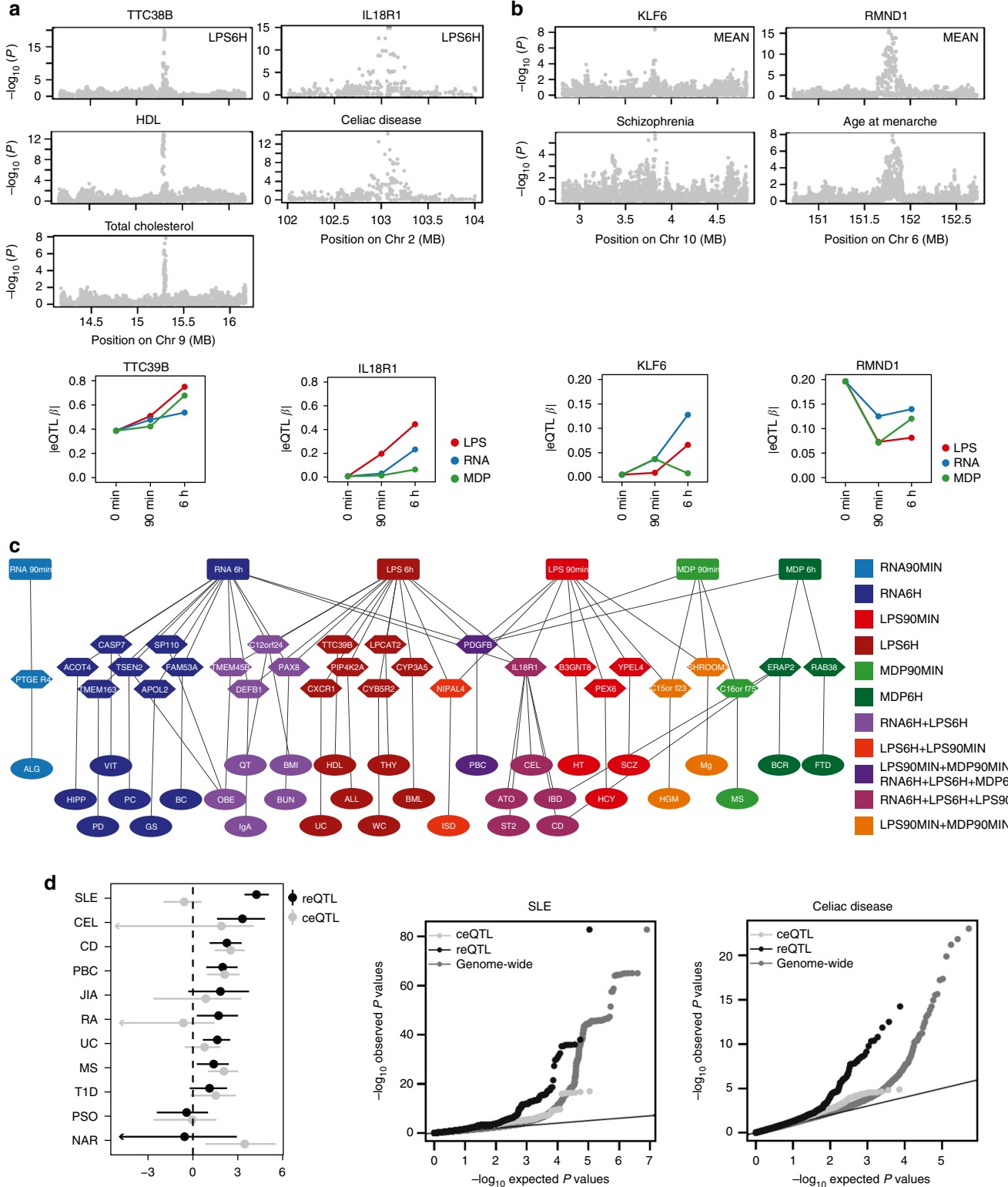

**Fig. 4** Immune response modifies genetic associations to disease. **a** Manhattan plots of eQTL (*top panels*) and disease (*middle panels*) *p*-values in colocalized loci. The *bottom panels* show the dynamics of corresponding eQTL effect sizes in different conditions. **b** Two additional GWAS loci colocalize when the mean of gene expression across all seven conditions is used to map eQTLs (see Methods section). **c** Overlap of GWAS SNPs that are in high LD ($r^2 > 0.8$) with reQTLs in monocytes with disease phenotypes connected to reQTL genes and corresponding immune stimulations. Supplementary Data 4 for trait abbreviations. **d** Genome-wide enrichment of reQTL and ceQTL associations in autoimmune GWAS with 95% confidence intervals (*left panel*), and Quantile–quantile (Q–Q) plots for SLE (*middle panel*) and Celiac disease (*right panel*). Supplementary Fig. 10b for additional Q–Q plots, and Supplementary Fig. 11 and Supplementary Fig. 12 for results of non-autoimmune traits

IRF transcription factors, chemokines, and chemokine receptors (Supplementary Fig. 6b, c, d), demonstrating the relevance of genetic interindividual variation in the innate immune system.

Next, to analyze treatment and time point specificity of reQTLs, we performed pairwise comparisons of regression slopes across treatments and time points, respectively. This revealed that 13–51% of reQTLs were treatment-specific when compared with the other two stimuli of the same time point, with marked differences depending on which stimulus-pair was tested (Fig. 2b). We also observed a large proportion of time point-specific reQTLs (32–64%) suggesting a highly dynamic genetic regulation in immune response (Fig. 2c). Of note, the number of identified reQTLs per condition, as well as time point- and stimulus-specific reQTLs, were correlated with the number of differentially expressed genes (Supplementary Fig. 7a, b). Thus, differential expression analysis in a small number of samples can be used to select the conditions that maximize reQTL discovery in a population-scale study.

To obtain better insight into the dynamic link between reQTLs and differential expression upon immune stimulation, we classified reQTLs into those with early transient, late, and prolonged effects (see Methods section). We find that active reQTLs, reQTLs that are absent at baseline and active under stimulation, are more common and have higher effect sizes than suppressive reQTLs, reQTLs that lose their baseline eQTL effect under stimulation (Fig. 2d, Supplementary Fig. 7c). Interestingly, active reQTLs are typically more dynamic showing early transient or late effects, whereas suppressive reQTLs are more often prolonged, extending over both time points. Next, we analyzed whether the temporal dynamics of reQTLs correspond to the dynamics of differential expression. A highly congruent pattern would indicate a major role of genetic interindividual variation in reQTL genes across the gene's temporal response to a stimulus, whereas divergent patterns could suggest recruitment of additional expression response mechanisms independent of the regulatory effect of the reQTL variant. The proportion of reQTL genes with congruent differential expression ranged between 30 and 87% for different classes of dynamic reQTLs (Fig. 2d, Supplementary Fig. 7d, Methods section) with significant enrichment of congruent pattern in 4 out of 6 groups ($p < 0.05$ in Fisher's exact test of each group vs all others). This indicates that reQTLs are relevant regulators of differential expression but additional regulatory mechanisms are involved in shaping the transcriptional response of reQTL genes. Altogether, our analysis of temporal reQTLs sheds light on mechanisms of the highly dynamic immune response, and the role of genetic variants in it.

**Functional mechanisms and evolution of reQTL variants.** To further characterize the genetic variants underlying the total of 417 reQTLs across all treatment conditions, we defined a set of 677 constant eQTLs (ceQTLs). These ceQTLs displayed no change in regression slope across all conditions (nominal $p > 0.05$) (Fig. 3a, Supplementary Fig. 8a) and genes with a ceQTL showed GO enrichment predominantly in metabolic processes (Supplementary Fig. 8b). Functional annotation enrichment and fine mapping analyses by fgwas[12] revealed that reQTLs were more enriched in promoter-flanking regions, CTCF binding sites and enhancer regions, while constant eQTLs were more common in promoter regions, 3′ and 5′ untranslated regions, and regions downstream of transcription start sites (Fig. 3b, Supplementary Fig. 8c). While reQTL enrichment has been previously described for some transcription factors[10, 11], and annotations of condition-specific epigenomic marks and tissue-specific eQTLs have been described[13, 14],

our results are to our knowledge the first demonstration of environmentally responsive eQTLs being enriched in distal *cis*-regulatory elements.

Given that the innate immune system is the first line of defense in the early interaction between the host and the microbe, we asked whether selective pressures that are exerted by microorganisms on the host genome can be detected in reQTLs. Consistent with previous reports[10, 11], we detected a signal of increased positive selection in eQTLs, ceQTLs, and reQTLs using the integrated haplotype score[15] (iHS; permutation test $p < 10^{-4}$, Fig. 3c, *left panel*) and the singleton density score[16] (SDS; permutation test $p < 10^{-4}$, Fig. 3c, *right panel*), comparing each eQTL class to a genome-wide null set of variants matched for minor allele frequency (MAF) and linkage disequilibrium (LD). Next, we examined the direction of the effect of the derived allele, dividing reQTLs into two groups (Fig. 3d): (1) reQTLs where the derived allele causes an increase in response amplitude compared with the ancestral allele (e.g., ancestrally upregulated genes are further upregulated among derived allele carriers), and (2) reQTLs where the derived allele causes weakening or even silencing of immune response compared with the ancestral allele. Interestingly, across all treatments the reQTLs with stronger expression response by the derived allele were more common (binomial $p = 0.011$ across all conditions; Fig. 3e, Supplementary Fig. 9). This suggests an evolutionary trend toward enhanced immune response, which might reflect an arms race of the host immune system and invading pathogens.

**Immune response modifies genetic associations to disease.** Given the central role of inflammation in many diseases, we examined reQTLs as a potential mechanism underlying genetic associations to complex diseases, discovered by genome-wide association studies (GWAS). First, we identified individual GWAS loci that are likely to share a causal variant with an reQTL in the same locus. We used the coloc[17] method on summary statistics of 33 GWAS traits (Supplementary Data 3) and our reQTL data. This analysis provided four loci with strong evidence (PP3 + PP4 ≥ 0.90 and PP4/PP3 ≥ 3) of reQTLs sharing the same causal variant with a GWAS trait (Fig. 4a, b and Supplementary Data 3). In the chromosome 9 locus associated with HDL[18] and total cholesterol levels[18], the eQTL effect for TTC39B can be detected at baseline levels, but the increasing effect size upon immune stimulation indicates a possible novel immunological component of TTC39B's role in the etiology of atherosclerosis. In the IL18R1 locus associated with celiac disease[19] (Fig. 4a) and the KLF6 locus associated with schizophrenia[20] (Fig. 4b), the eQTL effects are only present under immune stimulation and would not be discovered in baseline monocytes. Conversely, in the RNMD1 locus associated with age at menarche[21], the baseline eQTL effect is diminished upon immune activation. As summary statistics are only available for the minority of GWAS traits, we also identified 29 reQTL genes for which the top variant is in high LD ($r^2 > 0.8$) with a disease-associated SNP listed in the GWAS catalog[22] (Fig. 4c, Supplementary Data 4), which may indicate shared causal variants, albeit with less certainty than coloc analysis. For ten of these reQTL genes the eQTL was absent under baseline condition ($p_{baseline} > 0.01$), including reQTL genes such as APOL2 potentially associated with glomerulosclerosis, PTGER4 with allergy, and PIP4K2A with acute lymphoblastic leukemia. These results do not exclude other possible mechanisms in other cell types or conditions, but the reQTL analysis discovers potential causal genes for individual GWAS loci with an effect that is potentially modified by infections.

Finally, to quantify the role of reQTLs in the genome-wide genetic architecture of different complex traits, we analyzed the

enrichment of reQTLs and ceQTLs in GWAS signals of eleven autoimmune traits (Supplementary Data 3) using fgwas (Fig. 4d, Supplementary Fig. 10a), confirmed by Q–Q plots (Fig. 4d, Supplementary Fig. 10b) analogous to Li et al.[23]. Interestingly, in seven out of eleven traits reQTLs had a significant enrichment, whereas ceQTLs were enriched in only three of these seven traits, and narcolepsy (NAR) was the only trait significantly enriched for ceQTLs but not for reQTLs. Most notably, systemic lupus erythematosus (SLE) GWAS signals[24] were very strongly and significantly enriched among reQTLs with no enrichment in ceQTLs, suggesting that the innate immune response to pathogens may be a particularly important environmental modifier of genetic predisposition to SLE, while possibly playing a smaller role in the genetic architecture of e.g. psoriasis and type 1 diabetes. Even though fgwas analysis for multiple sclerosis (MS) did not show stronger enrichment of reQTLs over ceQTLs, the inflation of reQTLs in the QQ plot of MS advocates the importance of immune response genes in the etiology of MS (Supplementary Fig. 10). While some non-autoimmune traits showed an eQTL enrichment, there was no significant differential enrichment between reQTLs and ceQTLs (Supplementary Fig. 11, Supplementary Fig. 12). These results indicate a substantial, disease-specific role of environmental interactions with microbial ligands in genetic risk to complex autoimmune diseases. While tissue specificity of molecular effects of GWAS variants is increasingly appreciated and analyzed[14], our results suggest that innate immune stimulation is a key cellular state to consider in future eQTL studies as well as in targeted functional follow-up of GWAS loci.

## Discussion

In this study, we analyzed the interindividual variability of immune response in activated monocytes and characterized genetic variants that influence the response to pathogen components. Unlike previous studies, we analyze reQTLs using various ligands under multiple time points, and provide a more comprehensive picture of the role of genetic variation in innate immunity. Our analysis sheds light on the dynamics of immune response and reQTLs, the genomic elements underlying *cis* eQTLs responding to environmental stimuli, the evolution of immune response, and the key role of immune activation as a modifier of genetic effects especially in autoimmune diseases.

Several important aspects of genetic regulatory variants affecting transcriptional immune response remain to be addressed by other studies. RNA-sequencing allows increased power and identification of splicing QTLs[10, 23, 25], and additional epigenomic assays can provide insight into genomic mechanisms of transcriptome response[26]. Increasing sample sizes would provide better power and allow exploration into rare *cis*-eQTL variants[27, 28] and comprehensive *trans* eQTL mapping. Finally, while our study includes more immune stimuli and time points than previous analyses, it is essential to further expand the number of conditions and cell types involved in innate and adaptive immunity in reQTL studies, and advance their joint analysis. The ImmunPop QTL browser that includes our data provides a step toward this direction.

Taken together, our comprehensive characterization of reQTLs provide novel insights into the genetic contribution to interindividual variability and its consequences on immune-mediated diseases. These results support a model where genetic risk for disease can sometimes be driven not by static and uniform malfunction but rather by failure to respond properly to an environmental stimulus. This emphasizes the importance of context-specific genetic regulation in human traits.

## Methods

**Pilot study**. To assess the dynamics of immune response in human monocytes, we measured mRNA expression over a detailed time course of 45 min, 90 min, 3 h, 6 h, 12 h and 24 h following stimulation with 200 ng/ml ultrapure LPS from Escherichia coli (Invivogen), 100 ng/ml L18-MDP (Invivogen) or 200 ng in vitro transcribed 5′-ppp-dsRNA transfected with Lipofectamine 2000. These microbial ligands target three distinct pattern recognition receptor families and were chosen to study a broad spectrum of innate immune response in human monocyte. Differential expression analysis showed that early response genes are well captured at 90 min after stimulation followed by a "second" wave of late response genes that plateaued between 6 and 24 h after stimulation. Based on this pilot study, we profiled mRNA expression at 90 min and 6 h after stimulation in the larger eQTL cohort.

**Sample collection and stimulation of CD14+ monocytes**. In total, 185 healthy male volunteers of German descent were recruited. The study was approved by the institutional review board of the University of Bonn and informed consent was obtained from all donors. All volunteers were between age 18 and 35 (mean 24). Peripheral blood mononuclear cells (PBMC) were obtained by Ficoll-Hypaque density gradient centrifugation of heparinized blood. Monocytes were isolated by MACS using CD14-microbeads (Miltenyi Biotec) according to the manufacturer's instructions. Cell purity was assessed by FACS analysis of cell-surface antigens with a FACS LSRII (BD Biosciences). Monocytes were stained with an antibody against CD14 (V450 Mouse Anti-Human CD14 clone MφP9, BD Biosciences, catalog number 560349, 1:50 dilution) and purity was ≥ 95%. RPMI 1640 (Biochrom) supplemented with 10% heat-inactivated FCS (Invitrogen), 1.5 mM L-glutamine, 100 U/ml penicillin, 100 μg/ml streptomycin (all Sigma-Aldrich) and 10 ng/ml GM-CSF (ImmunoTools) was used to culture cells in 96-well round bottom wells at a density of 250,000 cells per well in 100 μl overnight. Cell viability after overnight incubation was > 85%. Cells of each volunteer were divided into subsets that were either left untreated or treated with 200 ng/ml ultrapure LPS from Escherichia coli (Invivogen), 100 ng/ml L18-MDP (Invivogen) or 200 ng in vitro transcribed 5′-ppp-dsRNA (IVT4) transfected with 0.5 μl Lipofectamine 2000 in a 50 μl reaction. Based on the pilot study described in Supplementary Fig. 1 and in the Methods section, cells were lysed in RLT reagent (Qiagen) after 90 min or 6 h and stored at −80℃. C-reactive protein (CRP) levels were measured to exclude samples with elevated CRP levels. After applying stringent quality control and clinical exclusion criteria (Non-smoker, no infection or vaccination 4 weeks prior to blood withdrawal, CRP < 2.5 mg/dl, monocyte purity ≥ 95%, monocyte survival > 85%), samples from 134 individuals were further processed.

**RNA extraction**. After stimulation cells were lysed and RNA was extracted using the AllPrep 96 DNA/RNA Kit (Qiagen). RNA quantity was determined using NanoDrop (PeqLab) and quality was assessed for a subset of samples using a Bioanalyzer (Agilent Technologies).

**Gene expression analysis**. RNA was amplified and biotinylated using Illumina TotalPrep-96 RNA Amplification Kit (Life Technologies) and gene expression analysis was quantified using Human HT-12 v4 Expression BeadChips (Illumina) comprising 47,231 probes. Expression profiles were quantile normalized, and only probes which showed a $p_{detection} < 0.01$ in at least 10 samples across all conditions were analyzed. Batch effects were removed using the R packages ComBat[29] and sva[30]. Probes with an interindividual standard deviation > 5 were set to NA. Probes found to map to multiple locations in the human genome or to non-autosomal chromosomes were not used. In addition, probes with SNPs that showed an eQTL effect on the respective gene were excluded, resulting in 18,988 probes (13,207 genes) for further statistical analyses.

To determine the number of differentially expressed genes, the probe with the best $p_{detection}$ across all conditions was used and differential expression ($\log_2$-fold change > 1, FDR 0.001) was computed using the linear modeling-based approach implemented in the Bioconductor limma package[31]. Genes differentially expressed in at least one condition were grouped into six distinct clusters corresponding to genes with similar response pattern using hierarchical clustering. Over representation of Gene Ontology terms in these clusters of differentially expressed genes was assessed using hypergeometric-based tests implemented in the R package GOstats[32]. Genes that were expressed in our monocyte data were used as background set in all enrichment analyses. Only enrichments significant at FDR of 0.05 are reported in Supplementary Data 1.

**DNA extraction**. Genomic DNA was extracted from 10 ml blood using Chemagic Magnetic Separation Module I (PerkinElmer Chemagen) according to the manufacturer's instructions. DNA was quantified by NanoDrop (PeqLab).

**DNA genotyping and imputation**. Genotyping was conducted on the Illumina's HumanOmniExpress BeadChips comprising 730,525 SNPs. After quality control ($p_{HWE} > 10^{-5}$, call rate > 98%, MAF > 5%), a total of 579,090 SNPs were available for analysis. Samples showing potential admixture within the multi-dimensional scaling (MDS) analysis were removed. All samples showed a call rate > 99%.

Genotypes were phased with SHAPEIT2[33] and imputed with IMPUTE2[34] in 5 Mb chunks against the 1000 genomes phase 1 v3 reference panel[35]. Sites with an

information score of less than 0.8 or significant departure from Hardy–Weinberg equilibrium ($p < 10^{-5}$) or MAF < 5% were excluded from further analysis. Genotype probabilities for all remaining sites were converted into dosage estimates.

**eQTL analysis**. As quantitative phenotypes, we used absolute expression values of untreated (baseline), LPS-treated (LPS), 5′-ppp-dsRNA (RNA), and MDP-treated (MDP) cells. Complete expression profiles of each of the seven conditions (baseline, LPS90min, LPS6h, RNA90min, RNA6h, MDP90min, MDP6h) were available for 134 donors. eQTL mapping was performed for SNPs located within 1 Mb of the gene expression probe using FastQTL[36]. Significance of the most highly associated variant per gene was estimated by adaptive permutation with the setting "--permute 100 10000". Permutation $p$-values obtained via beta approximation were used to access genome wide significance via Benjamini-Hochberg (FDR < 0.05). Downstream analyses were carried out in R. Network analysis of reQTL genes was performed using the STRING 10.0 database[37] selecting only interactions that were either experimentally validated or originated from curated databases.

**Replication of eQTLs**. We compared our results with two previous reQTL studies. For quantifying eQTL replication with a genome-wide study of monocyte eQTLs[9], we used Storey's qvalue R package[38]. The $\pi_1$ statistic considers the full distribution of association $p$-values (from 0 to 1) and computes their estimated $\pi_0$, the proportion of eQTLs that are truly null based on their distribution. Replication is reported as the quantity $\pi_1 = 1 - \pi_0$ that estimates the lower bound of the proportion of truly alternative eQTLs.

Lee et al.[8] used a targeted approach (415-gene signature) to identify eQTLs after LPS, Flu or IFNβ treatment in dendritic cells. $\pi_0$ could not be calculated using Lee et al. because less than 10% of eQTL genes in our data were represented in the 415 targeted genes, and thus replication was assessed by the proportion of our eQTLs with nominal significance ($p < 0.05$) in Lee et al.

**Detecting reQTLs by eQTL β-comparison**. In each condition, we first determined the best eQTL per gene (lead eSNP). Regression coefficient ($\beta$) and its variance ($\sigma^2$) of these eQTLs were calculated for all seven conditions using the linear model function summary(lm()) in R. We then tested if the regression coefficient of an eQTL was significantly different between two conditions in a $z$-test:

$$z = \frac{\beta_{baseline} - \beta_{stimulated}}{\sqrt{\sigma^2_{baseline} + \sigma^2_{stimulated}}}$$

Resulting $p$-values were corrected for multiple testing using Bonferroni correction ($p_{beta} < 0.05$). Previous reQTL studies[6, 8, 10] have used differential expression as a quantitative trait to identify reQTLs ($p_{diff}$). We calculated $p_{diff}$ for all reQTLs identified by $\beta$-comparison and used Spearman correlation as a measure of similarity.

To detect treatment specificity of reQTLs, we tested all significant reQTLs of one treatment (e.g., LPS90min) vs the other two treatments of the same time point (e.g., RNA90 and MDP90min) in two separate $z$-tests. A reQTL was treatment-specific if the Bonferroni-corrected $p$-value in the $z$-test was < 0.05. To detect time point specificity of reQTLs, for each treatment, we tested all significant reQTLs of one time point (e.g., LPS90min) vs the other time point (e.g. LPS6h) in a $z$-test. Time point-specific reQTLs were determined using Bonferroni-corrected $p$-values ($p < 0.05$). To compare reQTLs with eQTLs that are constitutively active (ceQTL), we defined ceQTLs as eQTLs with $p_{beta} > 0.05$ when testing each of the six stimulated conditions with the baseline condition.

**Characterizing dynamics of reQTLs**. To study the dynamics of reQTLs, we encoded as a binary call whether reQTLs had a significant eQTL $p$-value at each of the three time points or not (e.g., "0-1-0" codes for "not significant eQTL at 0 min —significant at 90 min—not significant at 6 h"). If a reQTL was shared between treatments, the treatment with the best $p$-value was used. This resulted in following groups: Transiently active ("0-1-0"), transiently suppressing ("1-0-1"), late active ("0-0-1"), late suppressing ("1-1-0"), prolonged active ("0-1-1"), and prolonged suppressing ("1-0-0") reQTLs. The average of absolute eQTL-$\beta$ and distribution of reQTL among these groups are shown in Fig. 2d (left panel). Of note, 83 reQTLs that were significant at all three time points ("1-1-1") but with significant changes of the eQTL effect size are not illustrated and were excluded from the following analysis.

To further examine if eQTL-$\beta$ and differential expression (DE) of the eQTL gene are congruent, DE between baseline and 90 min stimulation ($\Delta_{90\ min\text{-}baseline}$) and DE between 90 min and 6 h stimulation ($\Delta_{6\ h\text{-}90\ min}$) were calculated using limma and significant $\Delta_{90\ min\text{-}baseline}$ ($p < 0.01$) was encoded in binary (0;1) whereas significant $\Delta_{6\ h\text{-}90\ min}$ was encoded as "not significant" (0), "significant" (1), "significant, but opposite direction of $\Delta_{90\ min\text{-}baseline}$" (2). To determine the proportion of reQTL genes with congruent DE, we quantified for transiently active/suppressing reQTLs the proportion of reQTL genes with significant $\Delta_{90\ min\text{-}baseline}$ and significant $\Delta_{6\ h\text{-}90\ min}$ with opposite direction ("1-2"), for late active/suppressing reQTLs we quantified the proportion of reQTL genes with

not significant $\Delta_{90\ min\text{-}baseline}$ and significant $\Delta_{6\ h\text{-}90\ min}$ ("0-1") and for prolonged active/suppressing reQTLs we quantified the proportion of reQTL genes with significant $\Delta_{90\ min\text{-}baseline}$ and either not significant $\Delta_{6\ h\text{-}90\ min}$ (expression stays the same) or significant $\Delta_{6\ h\text{-}90\ min}$ with same direction (fold change increases, "1-0" or "1-1"). To test if the proportion of reQTL genes with congruent DE was significantly enriched in each group (e.g., 37 congruent of 71 transiently active reQTLs), we quantified the proportions of the same DE code (e.g., "1-2") in the remaining groups (late active/suppressing and prolonged active/suppressing) and tested the proportions using Fisher's exact test.

**Enrichment of functional annotations and fine mapping**. We used the fgwas[12] software to investigate the extent to which reQTLs and ceQTLs were enriched within specific annotation categories. Annotation information used by fgwas was derived from CADD variant consequence annotation[39] (14 annotations) and monocyte-specific annotations from Ensembl Regulatory build[40] (6 annotations). To identify the set of annotations that would best fit the model, we first tested each of the 20 annotations in a joint data set of reQTLs and ceQTLs including distance to TSS in the analysis. Sixteen annotations individually improved the model likelihood but as many of these annotations are correlated with one another we used a stepwise selection approach to identify a final best-fitting model that included 13 annotations asterisked in Fig. 3b. We then ran fgwas including these 13 annotations for reQTLs and ceQTLs separately to estimate enrichment parameters and output re-weighted summary statistics.

For each locus that contained at least one SNP with a posterior probability of association (PPA) > 0.3, we considered the SNP with the highest PPA from fgwas and tested the overlap of functional annotation sites of reQTL vs ceQTLs using Fisher's exact test. To increase power of reQTLs/ceQTLs overlapping functional annotation sites, we mapped eQTLs using the mean of gene expression across all seven conditions. Fgwas steps were repeated as described above. Estimated enrichment parameters showed similar results and indicate the robustness of our analysis (Supplementary Fig. 8b).

**Natural selection analysis**. We used two metrics, iHS and SDS, which detect signals of positive selection. The integrated haplotype score (iHS) measures the degree of extended haplotype homozygosity of the putatively selected allele over that of the putatively neutral allele[15]. iHS were calculated with the program selscan v1.1.0b[41] with default parameters. We defined high iHS values as |iHS| > 1.5 in the CEU population. Furthermore, we used the recently published singleton density score (SDS)[16], which detects very recent changes in allele frequencies from contemporary genome sequences. Publicly available SNP level SDS scores calculated from the UK10K Project reflect allele frequency changes during the past ~2000–3000 years in modern Britons, who are closely related to the German population[42]. We therefore applied these SDS scores to our cohort.

For each statistic (iHS, SDS), we determined the strongest signal of selection of all SNPs in high LD ($r^2 > 0.8$) with the best eQTL/ceQTL/reQTL SNP per gene. To assess significance, we then compared for each eQTL set the proportion of SNPs with |iHS| > 1.5 with the expected distribution obtained from re-sampling 10,000 sets of random SNPs matched for MAF and the number of SNPs in LD using the same parameters as described in Quach et al.[11] using bins of MAF of 0.05 and LD bins of 0–2, 3–5, 6–10, 11–20, 21–50, and > 50 SNPs with $r^2 > 0.8$). Similarly, for SDS, we compared the median of SDS scores of eQTLs/ceQTLs/reQTLs, to the expected distribution obtained from resampling 10,000 sets of random SNPs matched for MAF and LD patterns.

To determine the effect of the derived allele on the immune response, we tested the proportion of reQTLs where the derived allele causes an increase vs decrease in response amplitude compared to the ancestral allele (Fig. 3d). reQTLs with increased activity include both reQTLs where the derived allele amplifies the induction of a gene or amplifies the suppression of a gene, whereas reQTLs with decreased activity will either reduce the induction of a gene or reduce the suppression of a gene. Over representation of reQTLs with increased activity was evaluated using a binomial test.

**Colocalization analysis**. Colocalization analysis was conducted using the R package coloc[17]. The method requires summary statistics for each SNP, which were summarized in Pickrell et al.[43] or downloaded from ImmunoBase (http://www.immunobase.org) along with our eQTL data. A list of GWAS traits used in this analysis is provided in Supplementary Data 3. Coloc uses summary data from eQTL and GWAS studies in a Bayesian framework to identify GWAS signals that colocalize with eQTLs. We ran coloc using default parameter settings and a colocalization prior $p_{12} = 10^{-6}$. Coloc estimates posterior probability of association for either trait (PP0), association with gene expression (PP1), association with the trait (PP2), association with both phenotypes but distinct causal variants (PP3) and association with both phenotypes sharing the same causal variant (PP4). Regions with evidence for colocalization between gene expression and trait were defined as PP3 + PP4 ≥ 0.90 and PP4/PP3 ≥ 3 similar to what has been proposed by Guo et al.[44] and are illustrated in Fig. 4a.

As eQTL summary statistics in the coloc analysis, we used two approaches to maximize our discovery power. First, from each locus we used the summary statistics of the condition with the strongest $p$-value. This is expected to provide robust discovery even in highly condition-specific loci. Furthermore, we also ran

coloc with eQTLs mapped using the mean of gene expression across all seven conditions, which is expected to improve power when the eQTL signal is present in many conditions. All coloc results with PP3 + PP4 ≥ 0.90 are reported in Supplementary Data 3.

**Overlap between reQTLs and GWAS catalog**. To assess the overlap between reQTLs and trait-associated variants, we downloaded the NHGRI-EBI GWAS Catalog (version 1.0.1, downloaded 2016/06/14). A reported GWAS SNP was considered to coincide with an reQTL if the GWAS SNP was in high LD ($r^2 > 0.8$) with the lead eSNP per gene. A full list of these GWAS reQTLs is provided in Supplementary Data 4.

**Estimating the contribution of reQTLs on immune traits**. We used the fgwas[12] software to investigate the extent to which reQTLs and ceQTLs were enriched in risk loci of immune-mediated traits, following the approach of Li et al.[23]. A list of GWAS traits used in this analysis is provided in Supplementary Data 3. Due to the limited number of 417 reQTLs and 677 ceQTLs, we loosened the eQTL cutoffs for reQTLs and ceQTLs. For reQTLs, we considered all reQTLs that were significant after Benjamini-Hochberg FDR 5% correction (instead of Bonferroni correction), which resulted in 1128 reQTLs. For ceQTLs, we considered all ceQTLs with $p_{beta} > 0.005$ when testing each of the six stimulated conditions with the baseline condition, which resulted in 1165 ceQTLs. For both eQTLs, all associations with $p < 10^{-4}$ were used as input, and fgwas analysis was performed for reQTLs and ceQTLs separately. Of note, this analysis was robust to different eQTL association $p$-value cutoffs ($p < 10^{-4}$, $10^{-5}$, $10^{-6}$) suggesting that the enrichment is not simply due to the power of detection (Supplementary Fig. 10, Supplementary Fig. 11).

**Data availability**. Full summary statistics of the eQTL analysis and gene expression data are available in the ArrayExpress database (www.ebi.ac.uk/arrayexpress) under accession number E-MTAB-5631. In addition to results tables for all seven conditions provided in Supplementary Data 2, all eQTL results are available in the ImmunPop QTL browser (http://immunpop.com/kim/eQTL), which provides multiple interactive visualization and data exploration features for eQTLs.

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

## Acknowledgements

We thank all blood volunteers for participating to this study. We acknowledge our laboratory technicians and colleagues responsible for database management. S.K.-H. is supported by a research fellowship of the DFG. J.S. and V.H. received support for this work from the BONFOR research program, individual grant O-149.0094. J.S was supported by the NIH/DFG Research Career Transition Award. M.M.N. received support for this work from the Alfried Krupp von Bohlen und Halbach-Stiftung. V.H. is supported by the European Research Council (ERC-2014-CoG GENESIS 647858). M.M.N. and V.H. are members of the DFG funded Excellence Cluster ImmunoSensation. P.B. was supported by the SFB/Transregio 57 (TP25 and Q1) and an individual DFG grant (BO 3755/1-1). T.L. and P.M. were supported by the NIH grant R01MH106842.

T.L. was supported by the NIH grants UM1HG008901 and 1U24DK112331-01. T.L. and S.E.C were supported by the NIH contract HHSN2682010000029C.

## Author contributions

S.K.-H., J.S., and V.H. initiated the study. S.K.-H., B.P., P.M., Y.N., N.G., S.E.C., L.B.B., J.K.P., B.M.-M., J.S., V.H., and T.L. analyzed and interpreted the data. S.K.-H., J.B., M.B., V.K., E.B., N.F., and P.B. performed the molecular genetic experiments. S.K.-H., and M.B. characterized the volunteers and collected blood samples. S.K.-H., M.M.N., J.S., V.H., and T.L., prepared the manuscript, with feedback from the other authors.

## Additional information

**Competing interests:** The authors declare no competing financial interests.

