## [Peer Review File · Nature Communications]

Reviewers' Comments:

Reviewer #1:

Remarks to the Author:

This manuscript provides data on the eQTLs and reQTLs in human monocytes, extending this to stimuli beyond LPS, including MDP and dsRNA. Although the importance of response eQTLs in understanding the significance of variants with respect to disease pathogenesis is not a new concept, this study provides a wealth of observations that will be useful to the community. The enrichment of selection for derived alleles in enhanced response is interesting and perhaps particularly relevant to immune loci involved in autoimmunity. In this respect the striking the enrichment of reQTLs in SLE is intriguing. The authors might want to also comment on the skewing of the QQ plot in MS towards reQTLs, even though these are overall not enriched beyond ceQTLs in the MS associated genes.

While these observations are of interest, a major source of value for the scientific community is the ability to search online for these relationships. However, the website provided in the text (<http://immunopop/kim/eQTL>) did not always work for this reviewer. An alternative link at <http://132.219.138.157/kim/eQTL/> was available, but some of the search result did not result in the data shown in the paper. Several example, results for searches for CCL14 and CCL4L1 are shown below – these search results did not link to QTL data shown in the manuscript. In addition, the scale of the $-\log P$ on the Y axis shown for SNPs in search results sometime did not match the $-\log P$ values shown in the associated data. Finally, it would be useful to have the dots showing SNP data reveal the identity of rs marker when highlighted by hovering over these dots – this would facilitate searches for particular marker/gene QTLs.

Note: examples of web search results are shown in attached document

Reviewer #2:

Remarks to the Author:

In the manuscript, the authors explore the transcriptome of ex vivo monocytes taken from healthy male subjects and include a stimulation protocol with three agents that will activate interferon pathways and NF-KB by a variety of autoimmune disease-related pathways. The authors are well aware that characterising the transcriptome with arrays will provide far less information compared with RNA-Seq. Nevertheless, this is an important study, which highlights the issues of response eQTLs as being an essential component to the study of the transcriptome and its integration with genetic association data and epigenetics.

1 Why chose monocytes from males; males have a lower prevalence of almost all autoimmune diseases? In an ideal world, one would choose cells from both sexes, however, if funds were limiting sample size, then one would select cells from females as a preference.

2 It would be a insightful discussion point if the authors compared their results to those of Fairfax et al, who used arrays to examine transcriptome on resting and stimulated monocytes?

3. I did wonder whether the pathways implicated by the constant eQTL differed from those of the response dependent eQTL.

4. Were there any reciprocal changes in eQTLs in terms of direction of effect between stimulation protocols?

5 Did the age of the donors of the monocytes influence any of the responses measured?

Reviewer #3:

Remarks to the Author:

The authors aim to identify genetic regulatory effects that are modified by three different immune stimulations (LPS, MDP, ppp-dsRNA) in monocytes and determine how these correlate with GWAS

traits.

First of all, the manuscript is clearly written and contains all the necessary information in order to understand the study.

Moreover, the extensive time point analysis (45 min, 90 min, 3h, 6h, 12h, 24h stimulation) is very valuable for future stimulation studies. Especially, the observation that differential expression analysis in a small number of samples can be used to select the conditions that maximize novel reQTL discovery in a population-scale study.

A novel finding of this study is the observation that reQTLs are enriched in distal cis-regulatory elements. Furthermore, some novel reQTLs have been linked to multiple GWAS loci.

However, several findings are presented as being novel, but have been described in other publications before (see below). Based on this, the study should be better contextualized in the current literature and the incorrect classification of findings being novel should be corrected or the novel aspect should be further specified to be correct.

- "Response eQTLs are also enriched for recent positive selection with an evolutionary trend towards enhanced immune response.": this is a similar finding as in Quach et al., 2016 – Cell.
- "certain pattern recognition receptor (PRR) families such as NOD-like receptors have not been studied yet" : these have been studied before (but not specifically in monocytes). For example, Borrelia and Mycobacterium tuberculosis stimulations (activating NOD2) have been studied before in whole blood/PBMCs/macrophages (Janský et al., 2003 - Physiol. Res; Li et al., 2016 - Nat. Med.; Smeeckens et al., 2013 - Nat. Commun.; ter Horst et al., 2016 – Cell; Oosting et al., 2016 - Cell Host and Microbe).
- "thus far the dynamics of immune response have been only explored in LPS-treated cells. Unlike previous studies, we analyze various ligands under multiple time points" → Several earlier studies have explored the dynamics of the immune response after other stimulation than LPS (e.g. Amit et al., 2009 – Science; Li et al., 2016 - Nat. Med.; Janský et al., 2003 - Physiol. Res).
- "our comprehensive characterization of reQTLs provide novel insights into the genetic contribution to interindividual variability and its consequences on immune-mediated diseases. These results support a model where genetic risk for disease can sometimes be driven not by static and uniform malfunction but rather by failure to respond properly to an environmental stimulus." → Previous studies have shown similar findings for a wide variety of stimuli, but then related to cytokine-responses (e.g. Li et al. 2016 – Cell, Nature Medicine)

Despite the fact that several findings aren't novel, this study does provide a valuable, extensive analysis/comparison between multiple time points, multiple stimuli of reQTLs. And as such, reveals some more general features of these reQTLs (e.g. active reQTLs that are absent under baseline and active under stimulus are more common and have higher effect sizes than suppressive reQTLs where a baseline eQTL is lost under stimulus; active reQTLs are typically more dynamic with early transient or late effects, whereas suppressive reQTLs are more often prolonged, extending over both time points).

Several novel reQTLs are being mentioned in the text as an example, but it is unclear how many novel eQTLs/reQTLs have been identified in this study.

The observed findings were convincing and robust, as the eQTLs from conditions analyzed in previous studies had a high degree of replication (~75%) and several of their findings were confirmed using more than one approaches.

The main message of the paper "immune response eQTLs modulate autoimmune disease risk SNPs" is already known for some time (e.g. Barreiro 2012 – PNAS, Gat-Viks 2013 – Nat Biotechnol, Fairfax 2014 – Science). As such, I don't feel that the paper will influence thinking in the field.

I ascertained the statistical analyses that have been conducted, and believe these have been performed correctly.

Response to reviewers' comments

Reviewer #1

This manuscript provides data on the eQTLs and reQTLs in human monocytes, extending this to stimuli beyond LPS, including MDP and dsRNA. Although the importance of response eQTLs in understanding the significance of variants with respect to disease pathogenesis is not a new concept, this study provides a wealth of observations that will be useful to the community. The enrichment of selection for derived alleles in enhanced response is interesting and perhaps particularly relevant to immune loci involved in autoimmunity. In this respect the striking enrichment of reQTLs in SLE is intriguing.

1) *The authors might want to also comment on the skewing of the QQ plot in MS towards reQTLs, even though these are overall not enriched beyond ceQTLs in the MS associated genes.*

We thank the reviewer for pointing out the reQTL inflation signal in MS and agree that this finding should be highlighted in the main text. We have now included the following sentence in the current version of the manuscript (2nd paragraph on page 10):

“Even though fgwas analysis for multiple sclerosis (MS) did not show stronger enrichment of reQTLs over ceQTLs, the inflation of reQTLs in the QQ plot of MS advocates the importance of immune response genes in the etiology of MS (**Supplementary Fig. 10**).”

2) *A major source of value for the scientific community is the ability to search online for these relationships. However, the website provided in the text (<http://immunopop/kim/eQTL>) did not always work for this reviewer. An alternative link at <http://132.219.138.157/kim/eQTL> was available, but some of the search result did not result in the data shown in the paper. Several example, results for searches for CCL14 and CCL4L1 are shown below – these search results did not link to QTL data shown in the manuscript. In addition, the scale of the $-\log P$ on the Y axis shown for SNPs in search results sometime did not match the $-\log P$ values shown in the associated data. Finally, it would be useful to have the dots showing SNP data reveal the identity of rs marker when highlighted by hovering over these dots – this would facilitate searches for particular marker/gene QTLs.*

We highly appreciate that the reviewer tested the beta version of our immune reQTL browser and reported critical bugs. We have identified and eliminated all errors mentioned by the reviewer and the link (<http://immunopop.com/kim/eQTL>) should be fully functional now.

In more detail, different conversion tools to convert from Illumina expression probe IDs to HGNC gene symbols led to misassignment of eQTL results (such as CCL14 or CCL4L1).

This conversion step has been harmonized now between the eQTL browser and the manuscript and results of CCL14 and CCL4L1 can now be viewed in the eQTL browser.

P values of the regional association plots now match the tabular eQTL results that are shown in the “EQTL” section. Some genes were tagged by multiple expression probes that led to mismatches between the genome browser and the tabular data.

We thank the reviewer for the excellent idea to show SNP IDs when hovering over the association plot. We have implemented this function now in the current version of the genome browser, and by clicking on the dot of interest you will also see the corresponding boxplots in the section below the association plot. We are convinced this new implementation facilitates data exploration a lot and thank the reviewer again for the valuable comments.

Reviewer #2

In the manuscript, the authors explore the transcriptome of ex vivo monocytes taken from healthy male subjects and include a stimulation protocol with three agents that will activate interferon pathways and NF-KB by a variety of autoimmune disease-related pathways. The authors are well aware that characterising the transcriptome with arrays will provide far less information compared with RNA-Seq. Nevertheless, this is an important study, which highlights the issues of response eQTLs as being an essential component to the study of the transcriptome and its integration with genetic association data and epigenetics.

1) Why chose monocytes from males; males have a lower prevalence of almost all autoimmune diseases? In an ideal world, one would choose cells from both sexes, however, if funds were limiting sample size, then one would select cells from females as a preference.

We fully agree with the reviewer that a study design with both sexes would have been preferable. However, due to limited resources and the uncertainty of how much additional variance of the immune response is introduced by cyclical changes of sex hormones in women (Klein et al. 2016, PMID:27546235), we decided to include males only. Since several eQTL studies have now shown that you can account for factors such as sex or menstrual cycle day in the association model we will definitely consider both sexes in future studies.

2) It would be an insightful discussion point if the authors compared their results to those of Fairfax et al, who used arrays to examine transcriptome on resting and stimulated monocytes?

We are unsure if the reviewer noticed Supplementary Fig. 5a, which shows the comparison of our results with the results from Fairfax et al. (PMID: 24604202) and an additional study by Lee et al. (PMID: 24604203), where immune response eQTLs were identified in stimulated dendritic cells. The results show good replication rates,

consistently with earlier cis-eQTL studies, and this was already briefly mentioned in the text. Therefore, we assume that reviewer agrees that no further discussion on this topic is required.

3) *I did wonder whether the pathways implicated by the constant eQTL differed from those of the response dependent eQTL.*

We thank the reviewer for this important comment. We have added an additional supplementary figure (Supplementary Figure 8b) that includes the GO enrichment analysis of ceQTLs, which shows that ceQTL genes are enriched in metabolic pathways. We included following sentence in the current version of the manuscript (last paragraph on page 7):

“These ceQTLs displayed no change in regression slope across all conditions (nominal $p > 0.05$) (**Fig. 3a, Supplementary Fig. 8a**) and genes with a ceQTL showed GO enrichment predominantly in metabolic processes (**Supplementary Fig. 8b**).”

4) *Were there any reciprocal changes in eQTLs in terms of direction of effect between stimulation protocols?*

This is an interesting question since eQTLs with opposite effects under different stimulatory conditions might reveal regulatory variants with complex regulatory function (or LD artefacts). We therefore compared (r)eQTL β of either the two time points of the same stimulation (e.g. LPS90MIN vs LPS6H) or compared different treatment conditions against each other (e.g. LPS90MIN vs MDP90MIN, see the new Supplementary Fig. 5c). We were not able to identify any reQTLs with marked opposite effects in any of the tested treatment conditions. We included following sentence in the current version of the manuscript (1st paragraph on page 6):

“Of note, reQTLs with clearly opposite directional effect when comparing different treatment conditions were not observed (**Supplementary Fig. 5c**).”

5) *Did the age of the donors of the monocytes influence any of the responses measured?*

In our cohort, the age distribution was relatively narrow ranging between 18 and 35 with a mean of 24 (see figure below). We therefore did not expect any strong effects on our eQTLs. In addition, based on the analysis of bigger cohorts within the GTEx Consortium we have experienced that the effects of age on eQTLs are minor and would therefore not be detectable with the modest sample size of the current cohort.

Reviewer #3

The authors aim to identify genetic regulatory effects that are modified by three different immune stimulations (LPS, MDP, ppp-dsRNA) in monocytes and determine how these correlate with GWAS traits.

First of all, the manuscript is clearly written and contains all the necessary information in order to understand the study.

Moreover, the extensive time point analysis (45 min, 90 min, 3h, 6h, 12h, 24h stimulation) is very valuable for future stimulation studies. Especially, the observation that differential expression analysis in a small number of samples can be used to select the conditions that maximize novel reQTL discovery in a population-scale study.

A novel finding of this study is the observation that reQTLs are enriched in distal cis-regulatory elements. Furthermore, some novel reQTLs have been linked to multiple GWAS loci. However, several findings are presented as being novel, but have been described in other publications before (see below). Based on this, the study should be better contextualized in the current literature and the incorrect classification of findings being novel should be corrected or the novel aspect should be further specified to be correct.

We thank the reviewer for this important comment and apologize for being imprecise when reporting the novelty of our immune reQTL results. The confusion may have been caused by the fact that some of the sentences appeared to refer to immune response in general – a hugely diverse and widely studied field – rather than specifically to eQTL/reQTL studies, which is the focus of this work. Please find our comments to the specific examples below.

1) “Response eQTLs are also enriched for recent positive selection with an evolutionary trend towards enhanced immune response.”: this is a similar finding as in Quach et al., 2016 – Cell.

We agree with the reviewer that the enrichment of signals of natural selection in immune reQTLs have been thoroughly studied by Quach et al. and by another study (Nédélec et al.). We had already referred to their findings in the main text (2nd paragraph on page 8):

“Consistent with previous reports^{10,11}, we detected a signal of increased positive selection in eQTLs, ceQTLs, and reQTLs using the integrated haplotype score¹⁵ (iHS; permutation test $p < 10^{-4}$, **Fig. 3c**, left panel)”

The novelty of our finding is shown in the next figure (Fig. 3e and Supplementary Fig. 9) where we observed that reQTLs where the derived allele causes an increase in response amplitude were more common than reQTLs where the derived allele causes weakening of the immune response. This effect was seen across all treatment conditions and suggests an evolutionary trend towards enhanced immune response by the derived allele. To our knowledge this type of analysis of the direction of the derived allele effect on the immune response has not been reported before.

2.) *“certain pattern recognition receptor (PRR) families such as NOD-like receptors have not been studied yet”: these have been studied before (but not specifically in monocytes). For example, Borrelia and Mycobacterium tuberculosis stimulations (activating NOD2) have been studied before in whole blood/PBMCs/macrophages (Janský et al., 2003 - Physiol. Res; Li et al., 2016 - Nat. Med.; Smeeckens et al., 2013 - Nat. Commun.; ter Horst et al., 2016 – Cell; Oosting et al., 2016 - Cell Host and Microbe).*

We fully agree with the reviewer that many aspects of NLR activation including NOD2 have been studied by numerous previous studies. However, these studies were very different in their study design. The studies mentioned above primarily examined cytokine production at the protein level, and in the case of Jansky et al. and ter Horst et al. have not studied the genetic regulation of the immune system, which is the focus of our work. Additionally, all aforementioned studies have used whole organisms such as Borrelia and MTB that activate not only NOD2 but a plethora of other PRRs as well (Borrelia: TLR2/7/8/9, MTB: TLR1/2/4/9 and the inflammasome), which impedes the identification of NOD2-specific effects.

We have now clarified this in the Introduction to be more precise (2nd paragraph on page 3):

“For instance, reQTLs of certain pattern recognition receptor (PRR) families such as NOD-like receptors have not been studied with purified microbial ligands yet, and thus far the dynamics of immune reQTLs have been only explored in LPS-treated cells.”

We have now also included some of the above-mentioned literature as references in the Introduction (last sentence on page 2):

“Studying the genetic influence on immune response is complicated by the complexity of the immune system, which consists of many different cell types that respond to a plethora of signals, interact with each other and induce different effector functions under diverse kinetics¹⁻⁵.”

3) *“thus far the dynamics of immune response have been only explored in LPS-treated cells. Unlike previous studies, we analyze various ligands under multiple time points” à Several*

earlier studies have explored the dynamics of the immune response after other stimulation than LPS (e.g. Amit et al., 2009 – Science; Li et al., 2016 - Nat. Med.; Janský et al., 2003 - Physiol. Res).

We agree with the reviewer, and have revised the text to make clear that we are referring only to eQTL studies of the immune response and not immunological studies in general (Amit et al., Jansky et al.) or the genetic influence of cytokine production (Li et al.). We have now modified the sentences in the current manuscript to be more precise (2nd paragraph on page 3 and 11):

“For instance, reQTLs of certain pattern recognition receptor (PRR) families such as NOD-like receptors have not been studied with purified microbial ligands yet, and thus far the dynamics of immune reQTLs have been only explored in LPS-treated cells.”

“Unlike previous studies, we analyze reQTLs using various ligands under multiple time points, and provide a more comprehensive picture of the role of genetic variation in innate immunity.”

4) *“our comprehensive characterization of reQTLs provide novel insights into the genetic contribution to interindividual variability and its consequences on immune-mediated diseases. These results support a model where genetic risk for disease can sometimes be driven not by static and uniform malfunction but rather by failure to respond properly to an environmental stimulus.”* → *Previous studies have shown similar findings for a wide variety of stimuli, but then related to cytokine-responses (e.g. Li et al. 2016 – Cell, Nature Medicine)*

As mentioned by the reviewer both papers by Li et al. have deeply characterized the genetic influence on cytokine response upon multiple stimulations using ELISA. We agree that these studies greatly contribute to the general understanding of how genetics influences the immune system, and this sentence does not claim that we are the first ones to propose the model of response to stimulus being relevant for disease. However, we think that our GWAS results (colocalization and enrichment of immune reQTLs in GWAS loci) provide additional and important insights into the genetic role of immune-mediated diseases.

Despite the fact that several findings aren't novel, this study does provide a valuable, extensive analysis/comparison between multiple time points, multiple stimuli of reQTLs. And as such, reveals some more general features of these reQTLs (e.g. active reQTLs that are absent under baseline and active under stimulus are more common and have higher effect sizes than suppressive reQTLs where a baseline eQTL is lost under stimulus; active reQTLs are typically more dynamic with early transient or late effects, whereas suppressive reQTLs are more often prolonged, extending over both time points).

Several novel reQTLs are being mentioned in the text as an example, but it is unclear how many novel eQTLs/reQTLs have been identified in this study.

This is in fact a complicated question to answer, and thus any estimation of the number of novel eQTLs/reQTLs would be very difficult to interpret. There are two factors that contribute to this difficulty: 1) Full data sets of previous eQTL studies or even full lists of all their significant loci are often not readily available, and it is not even clear which studies should be included – monocyte studies certainly, but what about macrophage, PBMC, or whole blood studies? What about other tissues? 2) What is the statistical definition of novelty? Even though eQTL analysis produces binary calls of significance vs non-significance, the reality is gradual, and for example it is statistically inappropriate to estimate the overlap between two sets of eQTLs by simply tallying eQTLs that are significant in both. We also study conditions that have not been analyzed before – if e.g. a baseline monocyte eQTL, or and LPS-activated reQTLs has been reported before, is it a novel discovery if we show that it disappears under MDP stimulus?

Due to these challenges, and the fact that reporting such counts of novel eQTLs/reQTLs is not done in other genome-wide eQTL papers, we have chosen not to include this analysis in the manuscript. However, Supplementary Figure 5a provides analysis of replication of our findings, providing another type of a comparison to previous studies.

The observed findings were convincing and robust, as the eQTLs from conditions analyzed in previous studies had a high degree of replication (~75%) and several of their findings were confirmed using more than one approaches.

The main message of the paper “immune response eQTLs modulate autoimmune disease risk SNPs” is already known for some time (e.g. Barreiro 2012 – PNAS, Gat-Viks 2013 – Nat Biotechnol, Fairfax 2014 – Science). As such, I don’t feel that the paper will influence thinking in the field.

The concept of reQTLs modulating disease risk SNPs has indeed been studied in previous papers. However, our study provides additional insights that are truly novel: the eQTL analysis of individual loci shows novel reQTL signals that are of very high interest to people studying those specific loci. Furthermore, our statistically rigorous analysis comparing the genome-wide enrichment of response vs constant eQTLs provides new information of genetic architecture of several diseases, and for example the strong reQTL enrichment for lupus has not been reported before.

I ascertained the statistical analyses that have been conducted, and believe these have been performed correctly.

Reviewers' Comments:

Reviewer #1:

Remarks to the Author:

The website for these data is now substantially improved with the previous errors corrected. These results will be a valuable addition to the accumulating data on cell type- and stimulus- specific eQTLs/reQTLs.

Reviewer #2:

None

Reviewer #3:

Remarks to the Author:

The authors have adequately resolved the comments that I raised.